# Reducing Inert Materials for Optimal Cell–Cell and Cell–Matrix Interactions within Microphysiological Systems

**DOI:** 10.3390/biomimetics9050262

**Published:** 2024-04-25

**Authors:** Claudia Olaizola-Rodrigo, Héctor Castro-Abril, Ismael Perisé-Badía, Lara Pancorbo, Ignacio Ochoa, Rosa Monge, Sara Oliván

**Affiliations:** 1Tissue Microenvironment (TME) Lab, Aragón Institute of Engineering Research (I3A), University of Zaragoza, 50018 Zaragoza, Spain; colaizola@unizar.es (C.O.-R.); hcastro@unizar.es (H.C.-A.); iperise@unizar.es (I.P.-B.); soligar@unizar.es (S.O.); 2BEOnChip S.L., 50018 Zaragoza, Spain; larap@beonchip.com (L.P.); rmonge@beonchip.com (R.M.); 3Laboratorio de Biomiméticos, Universidad Nacional de Colombia, Bogotá 111321, Colombia; 4Institute for Health Research Aragón (IIS Aragón), 50009 Zaragoza, Spain; 5Centro de Investigación Biomédica en Red de Bioingeniería, Biomateriales y Nanomedicina, Instituto de Salud Carlos III, 28029 Madrid, Spain

**Keywords:** microfluidic devices, microphysiological systems (MPS), organ-on-a-chip (OoC), membranes, inert material, macro/micropores, mesh, migration, spheroids

## Abstract

In the pursuit of achieving a more realistic in vitro simulation of human biological tissues, microfluidics has emerged as a promising technology. Organ-on-a-chip (OoC) devices, a product of this technology, contain miniature tissues within microfluidic chips, aiming to closely mimic the in vivo environment. However, a notable drawback is the presence of inert material between compartments, hindering complete contact between biological tissues. Current membranes, often made of PDMS or plastic materials, prevent full interaction between cell types and nutrients. Furthermore, their non-physiological mechanical properties and composition may induce unexpected cell responses. Therefore, it is essential to minimize the contact area between cells and the inert materials while simultaneously maximizing the direct contact between cells and matrices in different compartments. The main objective of this work is to minimize inert materials within the microfluidic chip while preserving proper cellular distribution. Two microfluidic devices were designed, each with a specific focus on maximizing direct cell–matrix or cell–cell interactions. The first chip, designed to increase direct cell–cell interactions, incorporates a nylon mesh with regular pores of 150 microns. The second chip minimizes interference from inert materials, thereby aiming to increase direct cell–matrix contact. It features an inert membrane with optimized macropores of 1 mm of diameter for collagen hydrogel deposition. Biological validation of both devices has been conducted through the implementation of cell migration and cell-to-cell interaction assays, as well as the development of epithelia, from isolated cells or spheroids. This endeavor contributes to the advancement of microfluidic technology, aimed at enhancing the precision and biological relevance of in vitro simulations in pursuit of more biomimetic models.

## 1. Introduction

Over time, multiple efforts have been made to enhance the fidelity of in vitro simulation of human biological tissues. Among them, microphysiological systems (MPS) stand out as a promising alternative, mainly because they offer the possibility to perform large numbers of reproducible experiments with small amounts of reagents. This, in turn, is due to their low fabrication costs and their micrometric dimensions [1,2,3,4]. As this technology advances, organ-on-a-chip (OoC) approaches are rapidly emerging. These systems consist of engineered or natural miniature tissues cultured within microfluidic chips, possessing the potential to reduce the need for experimentation with animal models [5,6]. With their primary objective of replicating native physiological, histological and anatomical environments, OoC models are designed to include compartmentalization and spatial arrangement of cell or tissue-confining chambers, thus mimicking the highly organized structure of native tissues [7,8,9]. To accomplish this goal, the inclusion of inert materials in these devices, such as membranes, is necessary since they define the physical boundaries of each compartment.

They are fabricated with different materials, including polymers (polydimethylsiloxane (PDMS), polycarbonate (PC) or polyethylene terephthalate (PET)), and can feature micro- or nanoporous structures, enabling controlled interactions and separations within the device [10,11,12,13]. However, the introduction of these structures into the OoC models may impact the biological response of cells due to their non-physiological mechanical properties and composition [14]. Furthermore, depending on the nature of the intended biological simulation, the physical separation of different types of cells, created by such membranes, can create a significant limitation of the devices.

Cell–matrix and cell–cell interactions are crucial for the different cell types to preserve their physiological phenotype [15,16,17,18]. For example, in the skin, interactions between cells and the extracellular matrix (ECM) play a pivotal role in regulating normal homeostasis, aging, wound healing, and disease. Disruptions in integrin and ECM signaling are implicated in both tumor formation and fibrosis [19]. Similarly, cell-to-cell contact is critically significant for tissue development, maintenance of homeostasis, regeneration processes, and immune responses. Recent research emphasizes the substantial role of cell–cell interactions in the tumor microenvironment (TME), influencing tumor progression and metastasis [20]. Moreover, mammary epithelial cell’s interactions with their extracellular matrix play a key role in tissue development and functional processes of the tissue. Contemporary breast cancer treatment strategies take advantage of insights into the endocrine regulation of breast development and the evolving importance of stroma-epithelium interactions [21,22,23]. In the brain, the formation of neuronal connections during development is a vital process essential for the proper functioning of the central nervous system and regeneration in adulthood. The intricate process of axon extension and guidance involves interactions with signals from the extracellular milieu, including secreted factors, cells, axons, and extracellular matrix proteins, all contributing to shaping the wiring of the brain. The study of the spatiotemporal expression and mechanisms of these signals has improved our understanding of brain development and the potential treatment of central nervous system diseases. The importance of cell–cell and cell–matrix interactions in axon guidance is indeed great [24]. These examples, among numerous others, illustrate the crucial importance of maximizing the contact area between cells and matrices for a simulation that closely approximates the complexity of the human tissues.

Based on the above, the main objective of this study consisted of minimizing the presence of inert materials within the microfluidic device while ensuring appropriate cellular distribution. To achieve this goal, we developed two microfluidic devices by modifying the membranes separating the compartments of our designs. In the first one, denominated the Mesh device, a nylon mesh with regular pores of 150 µm was integrated into our device. This membrane decreased the amount of inert material by 55% of the overall area, favoring the direct contact between cells. The manufacturing methodology for this strategy also allowed for the incorporation of meshes with varying weights (regular pores of 250 µm or 363 µm) to achieve larger contact areas if necessary. In turn, the second design integrated an inert COC-Flex membrane with a 4 mm diameter, accommodating three macropores, each measuring 1 mm. This strategy was denominated as the Macropore device and favored the direct interaction between cells and scaffolds. For this device, the direct contact area amounts to 100%, eliminating the presence of inert materials in the contact zone. To validate the devices, we performed permeability and diffusion assays and compared their performance against chips with traditional polycarbonate (PC) membranes integrated with a 5% porosity [25]. Lastly, we present three possible biological applications with our designs. In the first one, the generation of a human epithelial layer with the Mesh device. In the second one, migration models of glioblastoma multiforme (GBM) throughout a collagen hydrogel layer are shown by using both the Mesh and Macropore devices. Finally, the third model presents, by using the Macropore device, the generation of another epithelial monolayer, with its connective tissue and native stromal cells. Our results show that inert materials can be removed from the chips to enhance direct cells and matrix contact without compromising the compartmentalization of the designs.

## 2. Materials and Methods

### 2.1. Microfluidic Device Fabrication and Validation

Both devices were designed using AutoCAD 24.0 (2D draws) and SolidEdge (3D render) software. They are manufactured from cyclic olefin polymers (COP) and copolymers (COC). The primary manufacturing method employed is thermocompression molding.

#### 2.1.1. Mesh Device

The platform consists of a COP (ZEONOR 1420R) injection piece, a nylon membrane of 150 μm for compartmentalizing the chip and replacing the standard membranes, and COC (TEKNIFLEX^®^ COC 100 E) layers that incorporate the channel and well profiles along with a COP base (ZEONOR 1420R film) (Figure 1).

The fabrication process begins with the creation of the channels (1.5 mm width and 200 µm height) and the base of the device using the cutting plotter FC8600-60 Wide Vinyl Cutter, Graphtec (Figure 1B). Subsequently, upon acquiring all the materials for the device, the thermocompression process commences. This process is carried out using the hot embosser Digital Combo Heat Press, Geo Knight. Initially, the channels are bonded to the base (100 °C, 40 s). Following this, the mesh is carefully positioned in the injection piece, and, finally, the ultimate bonding occurs at the same temperature of 100 °C for a period of 60 s. The completed device consists of two distinct wells, with a nylon mesh segregating them from the lower channels, as illustrated in Figure 1.

#### 2.1.2. Macropore Device

This chip is composed of cyclic olefin polymers (COP) and copolymers (COC). Following the scheme displayed in Figure 2A the top section of the device consists of an injection molded piece of COP (ZEONOR 1420R) which includes the inlets, outlets, and wells of the device. Universal screw connectors (1/4-28) can be inserted into these 4 inlets and outlets. The subsequent layers formed from (TEKNIFLEX^®^ COC 100), constitute the lower part of the wells, the membrane, and the channels. Finally, the base of the device is made of a layer of COP (ZEONOR 1420R film).

The manufacturing process begins with the production of the top part of the wells (ID = 5 mm, OD = 7 mm, h = 2 mm) using 3D printing technology (Asiga MAX UV) with UV curing resin (pro3dure’s audioprint^®^ GR-10). Subsequently, the injection piece is drilled using the OP-TIDRILL B13 BASIC machine with a 4 mm bit. A plotter cutting machine is used to make the geometries of all the layers: the bottom part of the wells (4 mm diameter and 388 µm height), membrane holes (1 mm diameter and 100 µm height) and channels (1.5 mm width and 300 µm height). Following this, a two-step thermocompression process is conducted through hot embossing to assemble the pieces. Initially, the layers will be bonded together (100 °C, 60 s) and subsequently, all these layers will be bonded to the injection molded part (100 °C, 40 s). Finally, the 3D printed wells are bonded to the injection piece with the aforementioned resin (60 s UVO). During this final stage, unpolymerized resin is applied along the periphery of the 3D wells and carefully positioned onto the injection-molded component encasing the injection wells. The resulting device incorporates two separate wells, each with a base featuring the holes that connect with the lower channels (Figure 2C).

To validate the device before initiating any biological experiment, the holes were filled with collagen to assess their precise confinement (Figure 3). The hydrogel mixture was prepared according to the required collagen concentration (4 mg mL^−1^). In this case, 50 μL of gel mixture was prepared by mixing the reagents in the following order: 0.61 μL of distilled water; 5 μL of DMEM 5X (Sigma-Aldrich, St. Louis, MI, USA; D5523); 0.47 μL of NaOH 1N (Sigma-Aldrich, St. Louis, MI, USA; 221465); 18.92 μL of collagen 10.57 mg mL^−1^; and finally, 25 μL of medium. To enhance the visualization of the hydrogel, green FluoSpheres (Invitrogen, Waltham, MA, USA; F8811) were introduced into the mixture. A 15 µL volume of collagen hydrogel was utilized to fill each hole, and the device was placed in an incubator (37 °C and 5% CO_2_) for 10 min for polymerization. Finally, a medium containing red FluoSpheres (Invitrogen, Waltham, MA, USA; F8809) was applied on top of each well and through the side channels to validate the device.

### 2.2. Cell Culture

In this study, a variety of cell lines were employed to perform the biological validation of both devices: Human colon carcinoma (HCT-116) from the American Type Culture Collection (ATCC, Gaithersburg, Maryland), human glioblastoma (U-87 MG) from (Sigma Aldrich, St. Louis, MI, USA; 89081402), human dermal fibroblasts (HDF from Gibco), cerebral microvascular endothelial cells (HBEC-5i, ATCC, Gaithersburg, Maryland; CRL-3245) and human colorectal adenocarcinoma cells (Caco-2, Sigma Aldrich, St. Louis, MI, USA; ECACC, 86010202).

The HCT-116 cell line was transfected with a lentiviral vector expressing green fluorescent protein (GFP), while the U-87 MG cell line was transfected with a lentiviral vector expressing cherry protein, following established protocols from previous studies [26]. HCT-116 and U87 cell lines were cultured in high glucose (4.5 g L^−1^, DMEM, Lonza, Basel, Switzerland; BE12-64F) supplemented with 10% fetal bovine serum (FBS, Sigma Aldrich, St. Louis, MI, USA; F7524), 2 mM Ultraglutamine (Lonza, Basel, Switzerland; 0MB074) and 1% penicillin/streptomycin (Lonza, Basel, Switzerland; 17-602E). This formulation is hereinafter referred to as DG10. Human dermal fibroblasts (HDF) were grown in DMEM with 1.0 g L^−1^ low-glucose (Lonza) culture media supplemented with 10% FBS, 1% penicillin/streptomycin and 1% Ultraglutamine. Caco-2 cell line was cultured with DG10, supplemented with 10% non-essential amino acids (NEAA) (Lonza, Basel, Switzerland; H3BE13-114E). HBEC-5i cells were cultured on vessels coated with 0.1% gelatin (Stemcell technologies, Saint Égrève, France; 07903) in DMEM:F12 medium (Gibco, Waltham, MA USA; 11320033), supplemented with 10% fetal bovine serum, 15 mM HEPES (Dutscher, Bernolsheim, France; 91L0180-11) and 40 µg mL^−1^ endothelial growth supplement (ECGS, Sigma Aldrich, St. Louis, MI, USA; E2759).

All cell cultures were maintained at 37 °C within a humidified incubator with 5% CO_2_. Cells were subcultured when they reached 80–90% confluence using 0.5 g L^−1^ Trypsin 1:250/0.2 g L^−1^ EDTA solution (Lonza, Basel, Switzerland; BE17-161E).

### 2.3. Spheroid Generation

HCT-116 and U-87 MG spheroids were prepared according to the liquid overlay technique inside round bottom 96 well plates (Sarstedt, Nümbrecht, Alemania). Prior to seeding, the plates were treated with a commercial anti-adherence solution (Stemcell technologies, Saint Égrève, France; 07010) to prevent cell attaching to the surface of the wells. Afterwards, a volume of 200 µL of a suspension comprising growth medium and appropriate number of cells (3000 for the HCT-116 and 1000 for the U-87 MG spheroids) was deposited inside each treated well. Thereafter, the plate was centrifuged at 1500 rpm for 10 min to ensure that all suspended cells remained at the bottom of the wells. Following this step, the aggregates were allowed to form for 24 h (HCT-116) and 72 h (U-87 MG) at 37 °C inside a humidified atmosphere with 5% CO_2_.

### 2.4. Collagen Hydrogel Preparation

3D culture experiments were performed using rat tail type I collagen hydrogels (Corning, Corning, NY, USA; 10224442). Hydrogels were prepared according to a previous protocol with a final collagen concentration of 4 mg mL^−1^ [27]. Herein, the culture medium used for the preparation of the hydrogels consisted of low-glucose Dulbecco’s Modified Eagle Medium (DMEM, Lonza, Basel, Switzerland; BE12-64F) supplemented with 1% penicillin/streptomycin and 2 mM Ultraglutamine. This formulation is herein after referred to as D0. Collagen hydrogels were prepared and deposited at 0 °C to prevent premature polymerization of the solution. Subsequently, the hydrogels were polymerized inside a humidified CO_2_ incubator at a constant temperature of 37 °C for 30 min.

### 2.5. Microfluidic Device Seeding

#### 2.5.1. Epithelium Generation

HCT-116 spheroids were directly seeded onto the mesh (Mesh device) to populate the area of the membrane and establish a cellular monolayer on it. To achieve this, 60 spheroids were positioned on the mesh, and the device was filled with DG10 covering both the interior of the well and the perfusion channel. Over the course of 11 days, the system was monitored daily until the epithelium was obtained. Following the establishment of the epithelium, 100,000 endothelial cells were seeded through the lower channel to mimic vascular endothelium.

#### 2.5.2. Migration Assays with U-87 Spheroids

U-87 MG spheroids were seeded inside the chips following the methodology described by Castro-Abril et al. for the “Constrained 2D experiment” [26]. Initially, a layer of collagen hydrogel solution (4 mg mL^−1^: 50 µL for Mesh and 20 µL for Macropore devices) was deposited into the wells of the microfluidic devices. Once polymerized, a mixture containing a single U87 MG spheroid (1000 initial cells) and collagen hydrogel solution (4 mg mL^−1^) was placed on top of the first layer and allowed to polymerize for the required time (30 min for Mesh and 15 min for Macropore devices) inside a humidified CO_2_ incubator at 37 °C. In the Mesh chip, the volume of the mixture was 50 µL, while in the Macropore model, it was 20 µL. In both cases, the perfusion channel was filled with DG10, while the top of the wells was filled with D0, recreating a nutrient gradient. Additionally, medium was refreshed every three days.

#### 2.5.3. Co-Culture Assays with Epithelial and Fibroblasts Cells

Human dermal fibroblasts (HDF) were trypsinized and stained with CellTracker™ Orange (CTO, Thermo Fisher Scientific, Waltham, MA, USA; CMTMR C2927,), at a concentration of 10 μM, diluted in culture medium without serum, and incubated at 37 °C for 20 min. Cells were seeded within a 2 mg mL^−1^ collagen hydrogel (200 cells/well), and the hydrogel was polymerized for 15 min at 37 °C. Subsequently, human colorectal adenocarcinoma cells (Caco-2) were trypsinized and stained with CellTracker™ Green (CTG: Thermo Fisher Scientific, Waltham, MA, USA; CMFDA C2927), using the same protocol as for CTO. Caco-2 cells were seeded on the surface of the hydrogel. HDF culture medium was added through the channels of the chip, beneath the hydrogel, and Caco-2 culture medium was added inside the well, above the hydrogel.

### 2.6. Membrane Permeability Evaluation

Experimental diffusion and permeability assays were performed to compare the permeability of fluorescein across the membranes featured in this document (150 μm pore nylon membrane integrated into the Mesh device, and the COC-Flex membrane consisting of 1 mm diameter pores integrated into the Macropore device) in presence or absence of cells, against a control polycarbonate (PC) membrane (8 μm diameter pore).

In all cases, the area of diffusion or permeability had a circular shape, with a radius of 0.5 mm in diameter and an area of 0.785 mm^2^. For the diffusion experiment, 3 replicates of each model (PC, nylon, and COC-Flex) were used. Collagen hydrogels were prepared according to the procedure described previously. Following preparation, 20 μL of collagen was deposited into each well and left to polymerize inside a humidified incubator at 37 °C, 5% CO_2_, and 95% air.

Subsequently, a volume of 36 μL of the donor solution (fluorescein dissolved in DG10) at a final concentration of 0.1 mg/mL was added on top of the layer of hydrogel. Samples (100 μL) were taken every 30 min from the perfusion channel of the devices for 3 h. After each measurement, the volume extracted from the channel was substituted with fresh medium. Between each measurement, the microdevices were placed on a rocker at a minimum speed at 37 °C in a humidified atmosphere of 5% CO_2_ 95% air.

Fluorescence was measured at each time point using the SynergyTM HT microplate reader (Biotek) with excitation and emission wavelengths of 440/30 nm and 530/25 nm, respectively. Serial dilutions of the standard curve were employed to convert measured fluorescence into equivalent concentrations of the samples obtained from the microfluidic devices during the assay [28,29].

Finally, the permeability coefficient was determined using the following formula [30]:$$P=\frac{\left(\frac{dC_{A}}{dt}\right)\cdot V_{A}}{A\cdot C_{D}}$$

In this equation, $\frac{dC_{A}}{dt}$ represents the slope derived from the equation obtained through cumulative permeability calculation (selected within the linear range),$\;C_{D}$ denotes the concentration of the donor, $V_{A}$ stands for the volume of the acceptor, and A represents the permeation surface.

### 2.7. Immunostaining

Upon completion of the epithelium generation in the Mesh device, the HCT-116 cell monolayer was fixed in 4% paraformaldehyde at room temperature (RT) for 30 min. Subsequently, cells were washed three times with PBS and permeabilized with 0.1% Triton X-100 in PBS for 15 min. The culture was subjected to three washes in abundant 0.05% Tween20 in PBS and further incubated in a blocking solution (3% BSA (A9418-506, Sigma-Aldrich) in PBS) for 3 h at RT. Following this, cells were washed three times with 0.5% BSA in PBS and labeled with an anti-ZO1 primary antibody (ZO-1 Polyclonal Antibody, Invitrogen) diluted 1:200 in 0.5% BSA in PBS. The incubation with the primary antibody was conducted overnight at RT. After the primary antibody incubation, cells were washed five times with 0.5% BSA (in PBS) before being incubated with a conjugated secondary antibody (Alexa Fluor 555, Invitrogen, A21428) diluted 1:500 in PBS for 2 h at RT. Finally, after three washes with 0.5% BSA) and then three times with PBS, nuclei staining with Hoechst (Hoechst 33342, Invitrogen) was performed by adding the solution to the cells at a final concentration of 1 ug mL^−1^, followed by incubation at RT for an additional 30 min.

### 2.8. Image Analysis

Confocal images of the devices were acquired using a Nikon Eclipse Ti-E equipped with a C1 modular confocal microscope while bright field and fluorescence images were acquired using a Leica DMi-8 (Thunder) microscope.

### 2.9. Graph Analysis

The graphs have been created using GraphPad 2.9.0 and analyzed using the Fiji software, through which various studies have been conducted. Firstly, the invaded area of the HCT monolayer was measured by calculating the occupied area over the total area, thus deriving the final percentage of the occupied area. Migration assays were analyzed by selecting the same area from the z stacks of both experiments (U-87 migration for Mesh and Macropore) and identifying the areas where migrating cells were located over time. This calculation yields the percentage of invaded area, i.e., the invaded area in each case divided by the total area of the selected block.

### 2.10. Statistical Analysis

A one-way ANOVA test was employed to examine the differences among the various membranes presented in this study. For this purpose, MATLAB (The MathWorks, Inc. Natick, MA, USA) and GraphPad Prism 8^®^ software (GraphPad Software Inc., USA, San Diego, CA, USA) were utilized. Note that a *p*-value below 0.05 indicates statistical significance, while a *p*-value above 0.05 indicates non-significance.

## 3. Results

### 3.1. Manufacturing of the Microfluidic Devices

The designs developed and validated demonstrated that biomimetic models with a direct contact area between the cavities of the fabricated microfluidic device can be achieved. Indeed, both configurations facilitate enhanced cell-to-cell interaction by replacing microporous membranes with either nylon mesh or COC macropores.

Concerning the Mesh device, the employed fabrication technique enables the integration of various mesh weights. Although the devices utilized in this study have been fabricated with a 150 μm pore mesh, this can be altered depending on other experimental demands as necessary. Throughout this project, devices featuring nylon meshes with pore sizes of 250 and 363 μm were also manufactured. The device’s versatility is enhanced by its ability to easily modify membrane properties and pore characteristics, thereby facilitating a broad range of experimental protocols from migration assays to monolayer formation.

Regarding the Macropore device, the design encompasses a wide volume range, with the macropore dimensions varying from 500 μm to 3 mm in diameter, tailored according to the specific application requirements. Additionally, the ability to adjust the well height, facilitated through 3D printing (the piece can be seen in Figure 2A), further expands the available range of volume selections. Ensuring complete contact between compartments across a substantial surface area enhances the feasibility of generating highly biomimetic models. In this instance, model validation entails seeding collagen gel with green fluospheres in the well, followed by the introduction of PBS containing red fluospheres into the upper zone upon polymerization (Figure 3). Confocal microscopy images demonstrate the precise confinement of the gel within the well and membrane pores. Additionally, examination of the z-axis images reveals the attainment of a uniformly flat gel layer, measuring 120 μm in thickness. In this case, a volume of 3 uL is seeded and allowed to undergo polymerization for a duration of 10 min.

### 3.2. Validation of the Microfluidic Devices

#### 3.2.1. Epithelium Generation

The Mesh device was customized to accommodate spheroids, organoids, or biopsies. In this model, human colon cancer spheroids were directly seeded onto the membrane to populate the entire mesh and establish a cellular monolayer (row B of Figure 4). The progression of the monolayer of cells generated from the spreading of HCT-116 spheroids over the surface is shown in Figure 4A,D. During the first 24 h (second micrograph of row A), HCT-116 cells gradually extended from the main mass and began encircling the nylon mesh. Subsequently, over the following hours, cells proceeded to occupy the vacant spaces between the nylon threads, resulting in the formation a cell layer. By the conclusion of the experiment (288th hour, last picture of Figure 4A and last point of Figure 4D), cells were able to colonize more than 95% of the total area of the well, generating a viable monolayer of cells. Regarding the stability of the cell unions inside the monolayer, immunostaining of the tight junctions of the HCT-116 monolayer with ZO-1 (labelled in red in the confocal micrograph in Figure 4C) revealed that the monolayer maintained its integrity in the colonized zones, even at the edges of the monolayer, with mosaic-like patterns typically seen in colon epithelium. Quantification of the area progression (Figure 4D) revealed that the spheroid cells invaded the well area in an almost linear fashion. From day 0 to day 12, there was a consistent increase in the invaded area, indicating a trend towards complete invasion of the well space, with coverage reaching up to 98.23% of the total area.

#### 3.2.2. Migration Assays

The results regarding the directed migration of U-87 MG cells in response to biochemical gradients, using the Mesh device are shown in Figure 5 and Figure 6. To enhance cell migration, a nutrient gradient was created by D0 for preparing the hydrogel and for filling the well. For the lower channel of the device, DG10 was used. In a first approach, we checked whether U-87 MG cells were able to reach the interface between the perfusion channel and the hydrogel in the Mesh device. As illustrated in Figure 5B,C, at 24 h, cells not only migrated along the interface of the two layers of hydrogels (confirmed by the bright-field micrograph) but, also began migrating towards the perfusion channel. Confocal imaging of the device indeed confirmed the presence of migrating cells that started invading the bottom layer of hydrogel. By 96 h, the cells had fully traversed the depth of this layer, reaching the nylon surface. 

We then replicated the migration assay without incorporating fluospheres into the bottom layer of hydrogel. Instead, we analyzed the interaction between glioblastoma and endothelial cells by seeding human brain microvascular endothelial cells (HBEC labeled in green) in the perfusion channel. This allowed us to investigate whether migrating U-87 MG cells (labeled in red) could come into contact with HBEC cells within a 24 h period. As observed in row C of Figure 6, migrating U-87 MG cells, were able to establish contact with the HBEC cells despite the short period.

Afterwards, we replicated the migration experiments in the Macropore device. Results are shown in Figure 7. As seen in the Figure, cells (highlighted in yellow) departed from the spheroid structure, migrating individually towards the channel as early as 24 h after the start of the experiment.

#### 3.2.3. Co-Culture Assays with Epithelial Cells and Fibroblasts

As stated above, we also co-cultured Caco-2 epithelial cells along with human dermal fibroblasts (HDF) was conducted in the Macropore device. Initially, HDF cells appeared as small dots within the hydrogel and making them challenging to locate in the microscope images. In addition, Caco-2 cells were suspended in the culture medium and were not stable on the interface of the hydrogel. However, after 24 h of seeding the device, HDF became visible, as they had begun to spread; and Caco-2 cells started to form a monolayer on the surface of the gel (Figure 8).

### 3.3. Permeability Assessment

Experiments with fluorescein were conducted to compare the performance of the membranes presented in this study against a standard PC membrane. After conducting two parallel experiments, one with a collagen layer and the other with the same layer with an added monolayer of HCT cells, the results depicted in Figure 9 were obtained. In Figure 9A, it was observed that over time, the collected concentration of fluorescein increased, with the Macropore devices exhibiting the highest concentration values, followed by the Mesh devices, and finally, the devices with PC membranes. Similar outcomes were observed upon analyzing the permeability coefficients of fluorescein in the presence of a monolayer HCT cells (Figure 9B). Specifically, by the end of the experiment, the values obtained for the Mesh or Macropore membranes were significantly greater than those obtained for the PC membrane, with respective *p*-values of 0.0059 and 0.0018.

## 4. Discussion

The compartmentalization in microfluidic devices is essential to replicate tissue architecture and achieve more biomimetic experimental models. In our endeavor to enhance intercompartmental contact, minimizing the presence of inert materials between them is paramount. Traditionally, for this purpose, polymeric materials such as polydimethylsiloxane (PDMS), polycarbonate (PC) or polyethylene terephthalate (PET) are used. It is also acknowledged that alternative materials such as poly(ε-caprolactone) (PCL) or poly(lactic acid) (PLA) are being employed [10].

PDMS, an elastomeric polymer, boasts several advantageous characteristics for biomedical applications. These include physiological inertness, exceptional resistance to biodegradation, biocompatibility, chemical stability, gas permeability, robust mechanical properties, outstanding optical transparency, and ease of fabrication through replica molding [31,32,33]. On the other hand, PC constitutes a distinct category of thermoplastic polymers [34]. They are recognized as inherently transparent thermoplastic materials distinguished by an amorphous structure. The molecular arrangement of polycarbonate imparts numerous advantageous characteristics to its physical properties, encompassing high stiffness, commendable thermal resistance, and elevated viscosity during material processing. Many examples of the use of membranes made from this material for microfluidic applications can be found, such as those for gut, blood–brain barrier and intestine [35,36,37,38,39]. Polyethylene terephthalate, commonly abbreviated as PET, stands out as one of the most widely utilized thermoplastic polymers in the market. It belongs to the polyester family, a broad category of polymers distinguished by the presence of ester functionalities within the macromolecular main chains [40,41]. Here as well, there are numerous applications where membranes of this material are employed such as kidney, liver and endothelium [42,43,44].

Another set of commonly used polymers in this field includes the aliphatic ones: poly(ε-caprolactone) (PCL) and poly(lactic acid) (PLA). PCL is a versatile biodegradable polyester extensively employed in biomaterial applications such as prosthetics, sutures, and drug delivery systems. PCL fibers, ranging from nanometers to millimeters in diameter, exhibit excellent characteristics like biocompatibility and three-dimensional porous structures, rendering them suitable for applications in drug delivery, absorbable sutures, and tissue engineering scaffolds [45]. On the other hand, PLA is a biocompatible thermoplastic derived from renewable sources. PLA is frequently used in manufacturing via 3D printing, laser cutting, or milling, allowing integration with electrodes or membranes for assembling complex microfluidic systems [46,47].

After thoroughly examining these materials used as membranes in microfluidic devices and the literature, membranes incorporating pores or holes larger than conventional micropores have not yet been established. Reviewing the existing literature, it is evident that the membranes currently utilized encompass a size spectrum down to 10 µm, dimensions that are comparatively smaller than those delineated in this research endeavor. Therefore, the technological approach outlined in this study represents a pioneering advancement in this field. Thanks to the manufacturing techniques developed for these devices, achieving the elimination of synthetic materials within the model, either partially as is the case with the Mesh or completely with the Macropore, is made possible. The intentional reduction in inert material, which acts as a physical barrier, provides significant advantages by promoting cell–cell or cell–matrix interactions and eliminating inert materials to achieve more biomimetic models in microphysiological systems. In the field of microfluidics, compartmentalized models are prevalent, typically characterized by material barriers separating individual compartments. While exceptions exist, such as certain devices facilitating total contact between channels, these instances do not align with the model configurations under discussion [29,48,49]. Notably, existing models are predominantly structured in a horizontal plane rather than a vertical orientation, thus diverging from the design principles inherent in our proposed models. This distinctive vertical orientation holds significant advantages, facilitating the creation of multiple layers of biological material, thereby enabling the simulation of complex multilayered structures such as the skin.

As outlined in the introduction, it has been demonstrated that most tissues necessitate close communication among all constituent elements to accurately replicate the physiological and pathophysiological conditions of the tissues or organs. In this regard, the biological validations conducted with the devices have demonstrated that the elimination of inert materials promotes cell–cell contact and facilitates optimal nutrient diffusion. A device resembling an organ-on-chip is considered more biomimetic due to its proximity to replicating real bodily functions. Consequently, the closer it can emulate the actual biological environment by minimizing inert materials, the more accurately it can be defined as biomimetic, as it allows for direct interaction between the device and bodily tissues, enhancing authenticity. The fact that the direct contact area between compartments can be increased from percentages of around 0.3, 2, or 15% (for PET and PC membranes found in the literature [10]) to percentages such as those of the Mesh device (45%) or Macropore (100%) is quite significant when discussing biomimetics. On the other hand, the percentage of direct contact area will alter the diffusion processes or permeability, as has been demonstrated. This suggests that specific membranes significantly influence the transport properties within the system. As noted in the permeability evaluations, there is a significant and noticeable difference when comparing the membranes presented in this study (Mesh and Macropore) to a commonly used PC membrane of 8 µm. This observation reinforces the concept of more direct contact and diffusion.

In the literature, cell monolayers are traditionally established by seeding individual cells on top of a solid membrane. While this approach may generate faster monolayers than our model, it also requires the presence of non-physiological materials that can alter cell mechanobiology [50,51]. In this case, this limitation is partially overcome by the presence of large pores in the nylon mesh, which reduces cell contact with inert material. Additionally, it provides minimal disturbance to the monolayer, as culture media can be added and replaced through the perfusion channels. Furthermore, it can be easily adapted to other epithelial cells, provided that they can form spheroids, by adjusting the initial size of the spheroids and the number of samples seeded inside the wells. This device, at the same time, facilitated the establishment of direct cell-to-cell interactions between the epithelial cells and the culture medium in the lower channel, or even with the endothelial cells seeded on the opposing side of the membrane. These features as a whole compound a complete model of epithelial tissue (in this case intestinal, but that can be simply modified by changing the cell line employed), present in different organs of the human body, such as the digestive tube itself, the trachea or the excretory system.

In addition, this work proposed a novel model for brain tumor cell migration towards the extracellular matrix, and, eventually, to the blood vessels. The design of these new microfluidic devices enables migrating cells to come into contact with the culture medium in the lower channel or with the endothelial cells seeded on the opposing side of the membrane when they reach the membrane frontline. U-87 MG is a cell line with a high tendency to migration, as reported in previous studies, and we have demonstrated in this study that when seeded forming a spheroid integrated in a collagen hydrogel, that migration does occur, and can be appreciated even 24 h after the seeding [52,53,54]. This model can provide insights into the interactions between tumor cells and the surrounding stroma, as well as into the mechanisms regarding glioblastoma progression and metastasis initiation. Furthermore, more solid tumors that lead to metastasis could be modelled with this device, since the pore present in the membrane permits the access of tumor cells into the blood circulation. Some examples could be breast cancer, lung cancer, or prostate, just by substituting the U-87 MG cell line by a tumor cell line proceeding from the type of cancer of interest.

As depicted in the images, the lack of materials in the compartments exposes the cells, allowing them to come into contact with other cells or directly interact with the extracellular matrix. This is particularly evident in the cell migration model with the co-culture of endothelial cells in the lower channel, mimicking a vessel. Moreover, both devices enable various experimental approaches, providing them with significant versatility when proposing OoC models. The utilization of three-dimensional structures such as spheroids, organoids, or small biopsies, along with hydrogels for reproducing the extracellular matrix, forms the foundation for replicating tissue architecture and obtaining more biomimetic models [55,56].

Furthermore, this study introduced a multicellular approach to model the system consisting of an epithelial monolayer and the supportive connective tissue beneath it. This was achieved by incorporating fibroblasts (the primary cells responsible for extracellular matrix generation in the human body) within a collagen hydrogel and seeding epithelial cells (specifically Caco-2, derived from human colorectal adenocarcinoma) on the surface of this matrix. Such a system is prevalent in various parts of the body, including the skin and the serous and mucous membranes.

Utilizing the aforementioned devices, an additional potential application involves the emulation of renal physiology through the establishment of bifurcated epithelial layers delineated by an interposed stromal matrix [57]. Another prospective application pertains to permeability assays, wherein the inert nature of the material minimally impacts or exerts negligible influence on the conducted assessments [58]. Considering potential advancements, the incorporation of these membranes into flexible microfluidic devices represents a notable progression. The integration of flexible models introduces an additional physical attribute conducive to enhancing model fidelity: mechanical stimulation. The development of flexible models featuring the COC membrane, as demonstrated by the Macropore device presented here, holds promise for realizing this vision. This aspect becomes particularly crucial when aiming to replicate organs or tissues that exhibit physiological movements, such as the heart or lungs [59,60,61].

## 5. Conclusions

The objective of achieving more biomimetic experimental models entails reducing the presence of inert materials within microfluidic devices. Two microfluidic devices have been developed to include membranes with pores large enough to facilitate substantial contact between cells and extracellular matrices. The first device, referred to as Mesh, features a built-in nylon membrane with regularly spaced pores measuring 150 μm in diameter. In contrast, the second device, named Macropore, incorporates a flexible COC membrane with macropores measuring 1 mm in diameter. Both devices have undergone biological validation through assays evaluating cell migration, epithelium generation and 3D co-cultures.

## Figures and Tables

**Figure 1 biomimetics-09-00262-f001:**
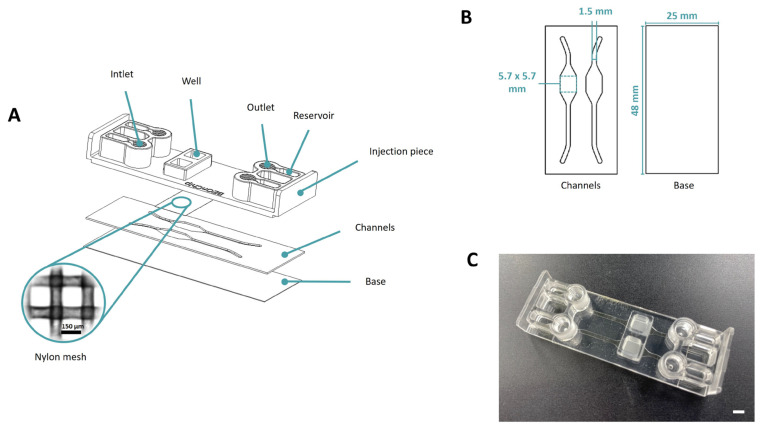
Design of Mesh device. (**A**) The device is composed of several parts: COP injection part, nylon membrane with 150 × 150 µm pores, COC Flex channels layer and COP base layer. (**B**) This section provides a visual representation of the design and dimensions of each layer, showcasing how they fit together to form the final device. (**C**) Final device. Scale bar: 5 mm.

**Figure 2 biomimetics-09-00262-f002:**
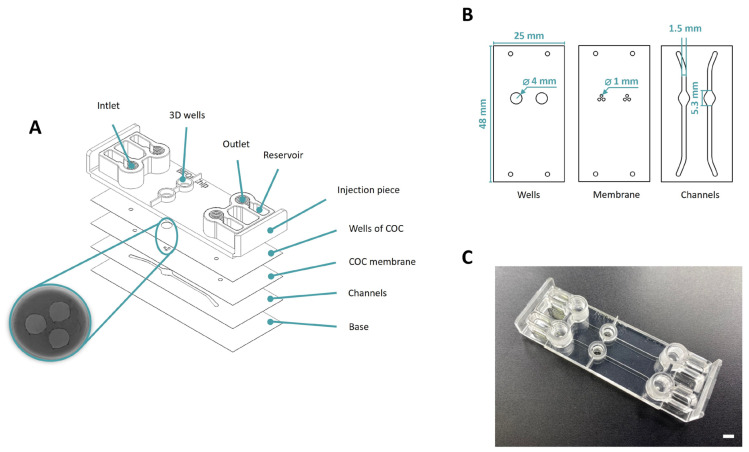
Design of Macropore device. (**A**) The device consists of several parts: COP injection part, COC Flex wells layer, COC Flex tri-pore membrane layer, COC Flex channels layer and COP base layer. (**B**) This section provides a visual representation of the design and dimensions of each layer, showcasing how they fit together to form the final device. (**C**) Final device. Scale bar: 5 mm.

**Figure 3 biomimetics-09-00262-f003:**
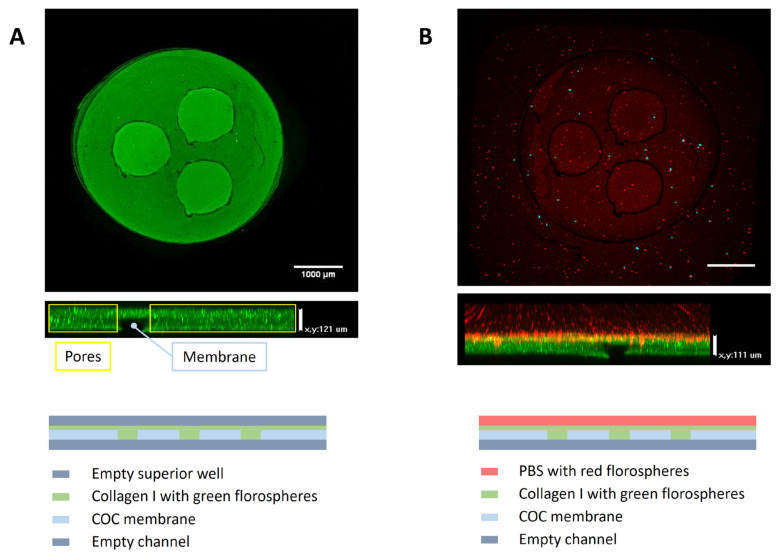
Validation of the Macropore device. (**A**) Top-down perspective and z-stack imaging of the recently inoculated well containing collagen gel infused with green fluospheres, with the seeding volume 3 µL. (**B**) Top-down view and z-stack imaging of the well containing the polymerized gel with freshly introduced PBS infused with red fluospheres. Scale bar: 1000 µm. At the bottom of the figure is a technical description of the schematic diagrams illustrating the structure and components of the well.

**Figure 4 biomimetics-09-00262-f004:**
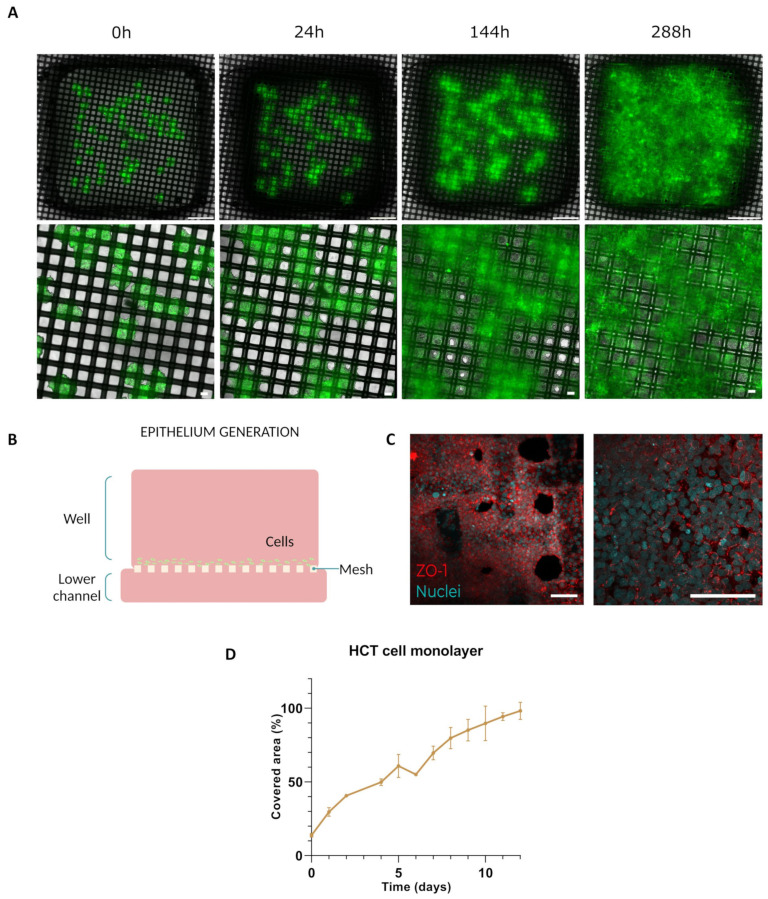
Epithelium generation. (**A**) Top-down perspective of the evolution of the monolayer at different time points (0, 24, 144, and 288 h). Insets represent details of cell spreading and colonization over the membrane. Scale bar top images: 1000 µm. Scale bar bottom images: 100 µm. (**B**) Schematic representation of the application. (**C**) Confocal imaging of the monolayer (20×) after immunolabelling of the tight junctions with ZO-1 (red) and nuclei counter labelling with Hoechst (cyan). The inset (40×, oil immersion) depicts the tight junctions of the monolayer at the selected zone. Scale bar: 100 µm. (**D**) Graph illustrating the percentage of the HCT cell monolayer’s area of occupancy on the nylon membrane over time.

**Figure 5 biomimetics-09-00262-f005:**
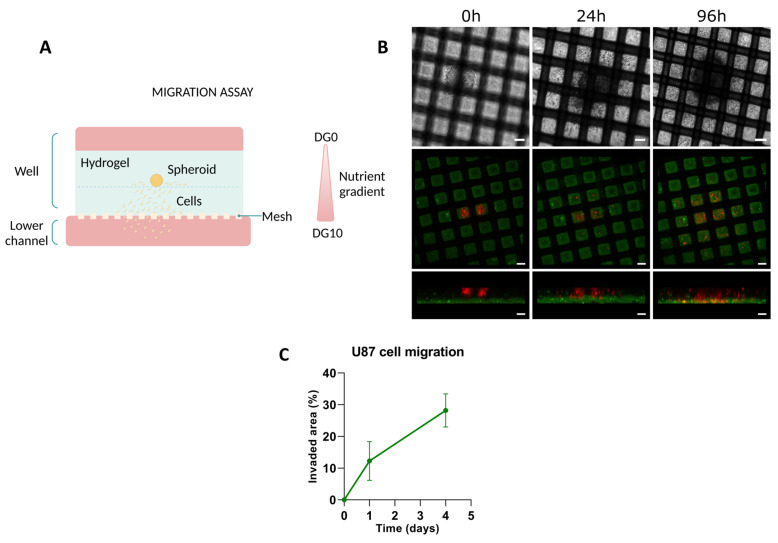
Migration assay on Mesh device. (**A**) Schematic representation of the application. (**B**) Time evolution of the invasive front of U-87 MG spheroids at different time points. The first row, corresponding to bright field images of the experiment, illustrates the invasion in the planar direction. The last two rows, corresponding to confocal images (10×) of the same experiment, depict the invasion front in the z-direction. Scale bar: 100 µm. (**C**) The progression of the invaded area as U-87 cells migrate throughout the studied 5 days.

**Figure 6 biomimetics-09-00262-f006:**
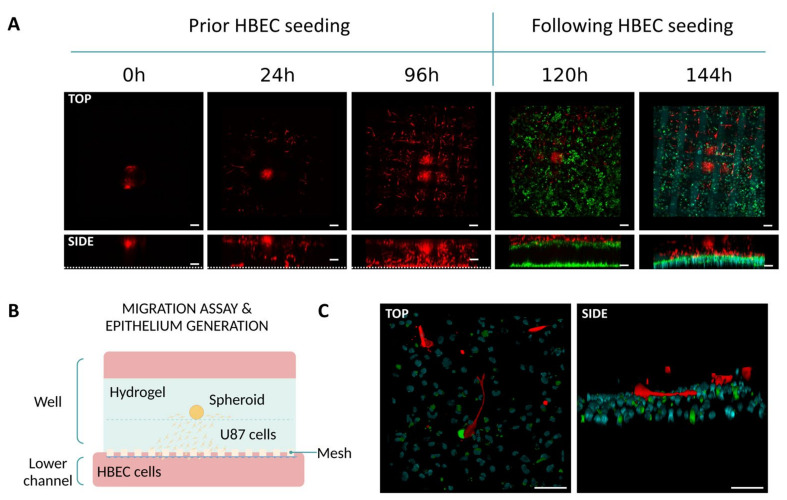
Migration assay and endothelium generation on Mesh device. (**A**) Confocal imaging captures the invasion of U-87 MG cells towards the perfusion channel over 96 h, alongside their interactions with endothelial HBEC cells during the 24 h period at 120 and 144 h. Scale bar: 100 µm. (**B**) Schematic representation of the application. (**C**) Z-stack reconstruction illustrating interactions between U-87 MG cells (in red) and endothelial cells (stained in green with their nuclei counterstained in cyan). Scale bar: 100 µm.

**Figure 7 biomimetics-09-00262-f007:**
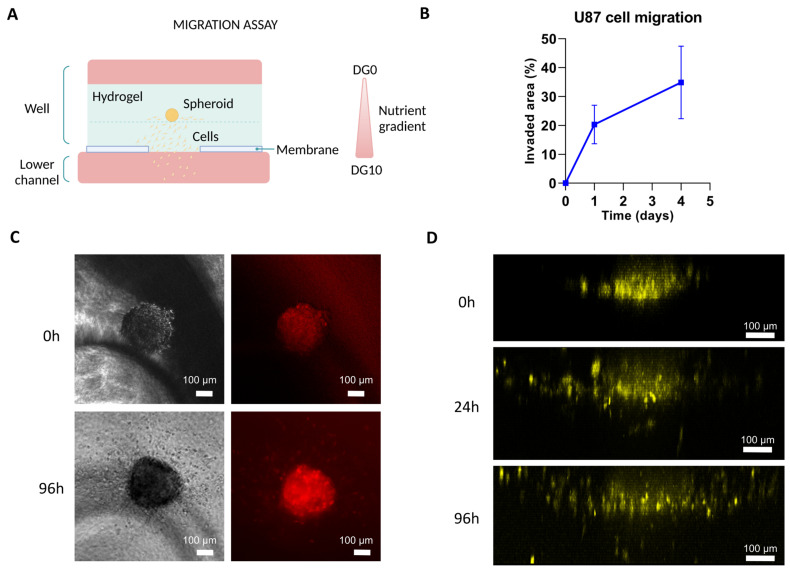
Migration assay on Macropore device. (**A**) Illustration depicting the experimental setup. (**B**) Analysis of the evolution of U-87 cell migration within the Macropore device over successive days, quantifying the invaded area as a percentage. (**C**) Images captured under phase contrast and fluorescence microscopy illustrating the development of a spheroid initially composed of 1000 U87 cells (top view), with the outline of the macropore visible just below. At 96 h, cell migration is already observable. (**D**) Confocal microscopy images (3D projection, lateral view) with cells artificially colored in yellow, highlighting their movement towards the channel. Scale bar: 100 μm.

**Figure 8 biomimetics-09-00262-f008:**
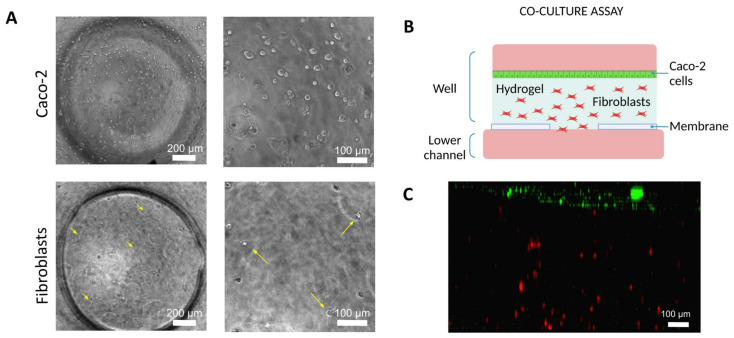
Co-culture assay with epithelial cells and fibroblasts on Macropore device. (**A**) Phase contrast images showing the Caco-2 cells (maximum height from the membrane) and HDF cells (intermedium height from the membrane). Yellow arrows have been added to mark their location. (**B**) Schematic representation of the application. (**C**) Confocal image (3D projection, lateral view). HDF are shown in red and Caco2 are shown in green. Scale bar: 100 μm.

**Figure 9 biomimetics-09-00262-f009:**
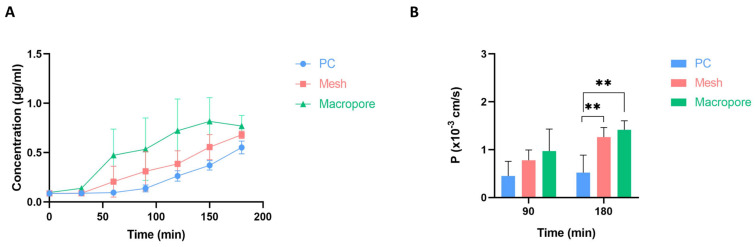
Permeability assessment. (**A**) Evolution of the quantity of fluorescein diffused through a 4 mg mL^−1^ collagen hydrogel in the chips with the studied membranes (PC, Mesh, and Macropore). (**B**) Permeability coefficient values at 90 and 180 min intervals for the three chips presented: the control chip featuring the PC membrane, the Mesh (chip containing the nylon membrane), and the Macropore. Data are represented as the mean  ±  SEM (** *p*  <  0.01; *n*  =  3).

## Data Availability

The original contributions presented in the study are included in the article, further inquiries can be directed to the corresponding authors.

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
