# Peer review of "Reducing Inert Materials for Optimal Cell–Cell and Cell–Matrix Interactions within Microphysiological Systems"

_biomimetics, 2024, doi:10.3390/biomimetics9050262_

Round 1

Reviewer 1 Report

Comments and Suggestions for Authors

This manuscript introduces two novel designs of microfluidic devices with the purpose of improving cell-to-cell and cell-to-matrix interactions. Specifically, these devices have different mesh permeabilities. Importantly, they are attempting to create more biomedical relevant models. However, this reviewer is concerned by the lack of quantification in the work. We think the device description is detailed with some nice preliminary experiments showing how they can be applied. However, we are concerned by data (or lack of it).

Major comments:

1.     The title of this manuscript connects higher cell-to-cell interaction with reduced interaction of the cell with  non-biological inert materials. However, there is no quantifiable comparison between the reduction of unspecific interactions and increase of cell-to-cell contact or cell migration. The visual data shows the interaction without providing any conclusive data on favorable choice between partial and complete elimination of inert materials.

2.     There is no quantification of the relative performance of the different designs. This would help the reader know which materials to choose if they have a specific assay in mind. Furthermore, there are no controls or even literature summary of conventional assays that show that this is better. 

3.     They make claims about tight junctions, but there is no data showing that they are actually tight. A simple experiment would be to do a dextran transport assay (or a TEER assay, but those can be difficult to compare in biomimetic assays). 

4.     There are migration assays. However, there is no quantification of how much is occurring. The pictures demonstrate that there is cell migration but we have no idea of what this activity is. 

Minor comments:

1.     A major share of the discussion contributes to discussing different inert materials utilized in the current field of research (literature review) instead of focusing on the interpreting experimental work and results.

2.     It would be helpful if the conclusion maybe discussed how this device fits into the field of biomimetic models. 

Author Response

We extend our sincere appreciation to the Editor and Reviewers for investing their time in evaluating our submitted manuscript. We have diligently addressed all raised concerns and integrated requisite revisions following the feedback received. In the ensuing sections, you will find the Reviewers' comments highlighted in black, while our responses are presented in italic format.

In response to the comments made by the reviewer 1:

Major comments:

  1. The title of this manuscript connects higher cell-to-cell interaction with reduced interaction of the cell with  non-biological inert materials. However, there is no quantifiable comparison between the reduction of unspecific interactions and increase of cell-to-cell contact or cell migration. The visual data shows the interaction without providing any conclusive data on favorable choice between partial and complete elimination of inert materials.

In response to the reviewer’s advice, we have quantified the migration of U87 cells within both the mesh and macropore devices, as well as the surface area occupied in the epithelial formation assay for HCT cells.

It is noteworthy that the development of these devices ensued after trials conducted with 8 μm pore PC membranes, wherein the cells were found to be retained within the membrane without migrating into the channel, thereby precluding direct contact between compartments. The experiments alluded to and depicted herein (Figure A) form part of an internal inquiry within the research group and, as such, are not suitable for inclusion in the publication. In the preceding experiment cited here, spheroids are observed by day 5, while cells remain adhered to the 8 µm pore PC membrane.

Figure A: Lateral perspectives of two spheroids captured via confocal microscopy are depicted. In each image, the upper boundary corresponds to the "equator" of the spheroid, while the lower boundary denotes the membrane of the chip. A) Spheroid 1 on day 0, B) Spheroid 1 on day 5, C) Spheroid 2 on day 0, and D) Spheroid 2 on day 5. Scale bar 200 µm.

On the other hand, concerning the membranes themselves, we have included in the manuscript details regarding the percentage of porosity in the membranes, specifically denoting the percentage of direct contact between the well and the channel within each respective device. Through this provided information, we demonstrate the significant disparity between the membranes commonly utilized in contemporary organ-on-chip devices and those presented within this study. This concept is introduced within the final part of the Introduction, commencing from line 94 onward.

“.... This membrane facilitates a direct intercompartmental contact spanning 45% of the overall area. The manufacturing methodology also allows for the incorporation of meshes with varying weights (regular pores of 250 µm or 363 µm) in order to achieve larger contact areas directly if necessary. In contrast, the second design integrates an inert membrane with a 4 mm diameter, accommodating three macropores, each measuring 1 mm. In this scenario, cells will be seeded within these macropores using a hydrogel. For this device, the direct contact area amounts to 100%, given that the gel is seeded within the 1 mm pore. Remarkably, the latest design eliminates the need for inert materials. To contextualize against membranes resembling those prevalent in contemporary literature, polycarbonate (PC) and Polyethylene (PET) membranes with porosities of 5% and 3%, respectively, are mentioned, as these are the ones utilized in our laboratory [25].”

[25]        “ipCELLCULTURE TM Track Etched Membrane,” vol. 32, no. 0, p. 226114, 2023.

  1. There is no quantification of the relative performance of the different designs. This would help the reader know which materials to choose if they have a specific assay in mind. Furthermore, there are no controls or even literature summary of conventional assays that show that this is better. 

Considering the reviewer's recommendation, an evaluation has been conducted to compare the two devices presented. The significance of the membranes presented is examined through a comparative analysis of migration experiments against other membrane types. This phenomenon is quantified in Figure B (in the manuscript we modified Figures 5 and 7 adding the graphs), where both studies demonstrate the migration of U87 cells from the spheroid over several days.

Figure B: Graphs for the migration assays of U87 spheroids for Mesh and Macropores devices.

The new section from Materials and Methods called “Graph analysis” explains the obtention and analysis of these graphs. The quantitative analysis is reflected in the graphs and the Results sections “Epithelium generation” and “Migration assays”, as well as throughout the Discussion paragraph.

Materials and Methods - 2.8. Graph analysis:

“The graphs have been created using GraphPad and analyzed using the Fiji software, through which various studies have been conducted. Firstly, the invaded area of the HCT monolayer was measured by calculating the occupied area over the total area, thus deriving the final percentage of the occupied area. Migration assays were analyzed by selecting the same area from the z stacks (lateral view) of both experiments (U-87 migration for Mesh and Macropores) and identifying the areas where migrating cells were located over time. This calculation yields the percentage of invaded area, i.e., the invaded area in each case divided by the total area of the selected block.”

Results - 3.2.1. Epithelium generation

“Quantification of the occupied area of the cell monolayer in the Mesh device (Figure 4D) illustrates that, overall, cells invade the unoccupied area of the device linearly. At the end of the experiment (day 12), cells occupied, on average, up to 98.23% of the total area of the well of the microfluidic device.”

This graph is added to Figure 4 (Figure 4D) of the manuscript:

Figure C: Graph for the monolayer formation along the days for the Mesh device.

Results - 3.2.2. Migration assays:

“Examination of the data reveals that on day 1, the invaded area for the Mesh device surpasses 10% (12,275% ± 6,118), whereas, for the Macropore device, it reaches 20% (20,313% ± 6,654). By day 4, the invaded area for the mesh device expands to 30% (28,2% ± 5,233), while for the macropore device, it approaches nearly 40% (34,880% ± 12,544). Consequently, it can be inferred that the migration rate in the Macropore device exceeds that of the Mesh device by approximately 10%. If we compare the two devices, the results are not statistically significant, even though at first glance it may appear that migration is greater in the Macropores device. ”

Discussion:

“Concerning the establishment of a confluent monolayer from a defined number of spheroids, as depicted in Figure 4, our findings show that by evenly distributing 60 spheroids, each comprising 3000 HCT-116 cells, across the surface of the Mesh microfluidic device, a monolayer was formed in approximately 12 days. This may be attributed, at least in part, to the high proliferation rates of this cell line (20-24 hours [48]) as well as the substantial number of spheroids seeded per well and the initial number of cells for their generation. In literature, cell monolayers are traditionally established by seeding individual cells on top of a solid membrane. While this approach may generate faster monolayers than our design, it also requires the presence of non-physiological materials that can alter cell mechanobiology [49]. In our case, this limitation is partially overcome by the presence of large pores in the nylon mesh that reduce cell contact with inert material. Additionally, it provides minimal disturbance to the monolayer, as culture media can be added and replaced through the perfusion channels. Furthermore, it can be easily adapted to other epithelial cells, provided that they can form spheroids, by tuning the initial size of the spheroids and the seeded number of samples inside the wells.

U-87 MG is a cell line with a high tendency to migration, as reported in previous studies [50]-[52], and we have demonstrated in this study that when seeded forming a spheroid integrated into a collagen hydrogel, migration does occur, and can be appreciated just 24 hours after the seeding. Thanks to the design of these new microfluidic devices, when the migrating cells reach the membrane frontline, they enter into contact with the culture medium in the lower channel, or with the endothelial cells seeded on the opposing side of the membrane, as it would happen at the beginning of a metastatic process.

As observed in the quantitative results, the difference in migration assays between the experiments conducted in the two devices is not statistically significant, despite a visible disparity of 10%. This finding is logical as the gel seeded in both devices remains consistent, as do the cells, which migrate similarly until they reach the membrane, after traversing the gel. Additionally, it is pertinent to mention that it is observed that migration in both devices is predominantly vertical, toward the perfusion channel.”

[48]         D. Ahmed et al., “Epigenetic and genetic features of 24 colon cancer cell lines” Oncogenesis. 2013, doi: 10.1038/oncsis.2013.35. PMID: 24042735; PMCID: PMC3816225.

[49]         MT. Lam et al., “Comparison of several attachment methods for human iPS, embryonic and adipose-derived stem cells for tissue engineering”. J Tissue Eng Regen Med., 2012, doi: 10.1002/term.1499

[50]         C. Metz et al., “Galectin-8 promotes migration and proliferation and prevents apoptosis in U87 glioblastoma cells” Biol Res, 2016, doi: 10.1186/s40659-016-0091-6

[51]         JT. Yang et al., “Propyl Gallate Exerts an Antimigration Effect on Temozolomide-Treated Malignant Glioma Cells through Inhibition of ROS and the NF-κB Pathway”  J Immunol Res. 2017, doi: 10.1155/2017/9489383

[52]         W. Huang et al., “Hypoxia enhances the migration and invasion of human glioblastoma U87 cells through PI3K/Akt/mTOR/HIF-1α pathway” Neuroreport. 2018, doi: 10.1097/WNR.0000000000001156

Alternatively, as per the reviewer's suggestion, the feasibility of conducting a permeability test has been contemplated. However, its execution demands a time investment surpassing our current availability. This is due to the necessity of manufacturing multiple devices, including one with a membrane designated as a control, synthesizing requisite compounds such as dextrans, conducting thorough assessments to determine their suitability based on varying applications, or exploring multiple options to observe permeability across different particle sizes.

  1. They make claims about tight junctions, but there is no data showing that they are actually tight. A simple experiment would be to do a dextran transport assay (or a TEER assay, but those can be difficult to compare in biomimetic assays).

As outlined in the manuscript, we conducted an immunodetection procedure targeting the characteristic tight junctions of epithelial cells, utilizing the ZO-1 antibody for labeling. In Figure 4 of the manuscript, this labeling is visually represented by the color red. We acknowledge the reviewer's observation that, by the end of the experiment, the monolayer had not fully formed. Assessing the integrity of the monolayer in its entirety would indeed be valuable; however, such an analysis necessitates conducting permeability studies, which would require additional time to execute. Nonetheless, the ZO-1 labeling serves as an indicator of the presence of tight junctions, as evidenced in the image below:

Figure D: Confocal imaging of the monolayer (20x) after immunolabelling of the tight junctions with ZO-1 (red) and nuclei counter labeling with Hoechst (cyan). Scale bar: 100 µm.

  1. There are migration assays. However, there is no quantification of how much is occurring. The pictures demonstrate that there is cell migration but we have no idea of what this activity is.

As per the recommendations provided in the reviewer's second response, we proceeded to quantify the migration assays for both the Mesh device and the Macropore. Figure B within this document displays the corresponding graphs for each device, highlighting the disparities in U87 cell migration. The quantitative aspect of migration is elucidated in the second point within this block of responses, and the manuscript integrates this discourse into sections “Epithelium generation” and “Migration assays” of Results and the Discussion section as explained and reported in answer 2 of this section of responses (reviewer 1).

Minor comments:

  1. A major share of the discussion contributes to discussing different inert materials utilized in the current field of research (literature review) instead of focusing on the interpreting experimental work and results.

Upon the recommendation of the reviewers, additional remarks regarding the experiments have been incorporated to enhance the comprehensiveness of the article (line 415 of the Discussion) as explained in question 2 of reviewer 1. The section discussing the available materials in the bibliography has also been condensed in the Discussion:

“PDMS, an elastomeric polymer, boasts several advantageous characteristics for biomedical applications. These include physiological inertness, exceptional resistance to biodegradation, biocompatibility, chemical stability, gas permeability, robust mechanical properties, outstanding optical transparency, and ease of fabrication through replica molding [28]–[30]. On the other hand, PC constitute a distinct category of thermoplastic polymers [31]. They are recognized as inherently transparent thermo-plastic materials distinguished by an amorphous structure. The molecular arrangement of polycarbonate imparts numerous advantageous characteristics to its physical properties, encompassing high stiffness, commendable thermal resistance, and elevated viscosity during material processing. Many examples of the use of membranes made from this material for microfluidic applications can be found, such as those for the gut, blood-brain barrier, and intestine among others [32]–[36]. Polyethylene terephthalate, commonly abbreviated as PET, stands out as one of the most widely utilized thermoplastic polymers in the market. It belongs to the polyester family, a broad category of polymers distinguished by the presence of ester functionalities within the macromolecular main chains [37], [38]. Here as well, there are numerous applications where mem-branes of this material are employed such as kidney, liver, and endothelium [39]–[41]. Another set of commonly used polymers in this field includes the aliphatic ones: Poly (ε-caprolactone) (PCL) and Poly (lactic acid) (PLA). PCL is a versatile biodegradable polyester extensively employed in biomaterial applications such as prosthetics, sutures, and drug delivery systems. PCL fibers, ranging from nanometers to millimeters in diameter, exhibit excellent characteristics like biocompatibility and three-dimensional porous structures, rendering them suitable for applications in drug delivery, absorbable sutures, and tissue engineering scaffolds [42]. On the other hand, PLA is a biocompatible thermoplastic derived from renewable sources. PLA is frequently used in manufacturing via 3D printing, laser cutting, or milling, allowing integration with electrodes or membranes for assembling complex microfluidic systems [43], [44].”

[All references were included in the initial manuscript]

  1. It would be helpful if the conclusion maybe discussed how this device fits into the field of biomimetic models. 

As recommended by the reviewer, this information has been added to the discussion (from line 492 onwards):

“A device resembling an organ-on-chip is considered more biomimetic due to its proximity to replicating real bodily functions. Consequently, the closer it can emulate the actual biological environment by minimizing inert materials, the more accurately it can be defined as biomimetic, as it allows for direct interaction between the device and bodily tissues, enhancing authenticity. The fact that the direct contact area between compartments can be increased from percentages of around 0.3, 2 or 15 % (for PET and PC membranes that we could find in the literature [10]) to percentages such as those of the Mesh device (45 %) or Macropore (100 %) is quite significant if we are talking about biomimetics.”

[The reference was included in the initial manuscript]

Figures A, B, C and D can be seen in the document attached to this response. 

Reviewer 2 Report

Comments and Suggestions for Authors

The authors have developed a fabric-based microfluidic chip for organ-on-chip applications. They have carried out a series of assays to test the capability of the system. As a technique for cost-effective biomimetic solutions for spheroids, the work is presented well. However, there is no single quantitative information like a plot in the paper. Authors are suggested to include them and discuss more.

Author Response

We extend our sincere appreciation to the Editor and Reviewers for investing their time in evaluating our submitted manuscript. We have diligently addressed all raised concerns and integrated requisite revisions following the feedback received. In the ensuing sections, you will find the Reviewers' comments highlighted in black, while our responses are presented in italic format.

In response to the comments made by the reviewer 2:

The authors have developed a fabric-based microfluidic chip for organ-on-chip applications. They have carried out a series of assays to test the capability of the system. As a technique for cost-effective biomimetic solutions for spheroids, the work is presented well. However, there is no single quantitative information like a plot in the paper. Authors are suggested to include them and discuss more.

Following the reviewers' suggestion, graphs have been included in the documents, they can be seen in Figures 4, 5, and 7. The interpretation and discussion of all of them appear in the Results (“Epithelium generation” and “Migration assays” points) and Discussion (from line 502 forward) sections.

Reviewer 3 Report

Comments and Suggestions for Authors

The authors suggests ways to improve and mimic cell to cell interactions within micro-environment. The objective was of achieving more biomimetic experimental models within the microfluidic devices. Two microfluidic devices have been developed to include membranes with pores large enough to facilitate substantial contact between cells and extracellular matrices. In fact, the work is interesting and useful to advance the field of microfluidic testing. However, the motivation to undertake this work and applications needs improvement. Following points should be incorporated in revised version of the manuscript:

1)       Did the author compare their device performance with any of the traditionally available polymeric materials such as (PDMS), polycarbonate (PC), or polyethylene terephthalate (PET)? Please report.

2)      Provide few more applications for the biomimetic system reported

Minor comments

1. Proof read the document to ensure grammatical errors

2. Sub sections can be organized in a better way

3. Line 53 cell-to cell sentence is not complete. Please check at other locations too.

Comments on the Quality of English Language

Improvemnt required. Please get it proof read

Author Response

We extend our sincere appreciation to the Editor and Reviewers for investing their time in evaluating our submitted manuscript. We have diligently addressed all raised concerns and integrated requisite revisions following the feedback received. In the ensuing sections, you will find the Reviewers' comments highlighted in black, while our responses are presented in italic format.

In response to the comments made by the reviewer 3:

The authors suggest ways to improve and mimic cell to cell interactions within the micro-environment. The objective was of achieving more biomimetic experimental models within the microfluidic devices. Two microfluidic devices have been developed to include membranes with pores large enough to facilitate substantial contact between cells and extracellular matrices. In fact, the work is interesting and useful to advance the field of microfluidic testing. However, the motivation to undertake this work and applications needs improvement. Following points should be incorporated in revised version of the manuscript:

1)       Did the author compare their device performance with any of the traditionally available polymeric materials such as (PDMS), polycarbonate (PC), or polyethylene terephthalate (PET)? Please report.

As discussed in reviewer 1's response 1, previously in our research group, we employed 8 μm pore size polycarbonate (PC) membranes for spheroid migration experiments. As depicted in Figure A, cells adhere to the membrane (no migration), prompting the development of the two devices presented in this study. While these assays were not conducted concurrently and are associated with another publication, hence not included in this work, we recognize the potential value of comparing with membranes cited by the reviewer from the literature. However, such comparison necessitates additional time beyond what was available for this review, as it would entail selecting appropriate membranes for these experiments, fabricating the devices, and conducting biological experiments over an extended duration.

2)      Provide few more applications for the biomimetic system reported

Thanks to the reviewer's suggestion, we have incorporated additional applications of the device beyond those previously mentioned (line 502 of the Discussion section):

“Firstly, epithelial tissue was modeled using the Mesh device. By seeding HCT-116 (human colon carcinoma) cells on the microfluidic device membrane, we could mimic the characteristics of the intestinal epithelium. To accomplish this, the utilization of an intermediate porous membrane, such as the one present in the Mesh device, allowed the formation of a monolayer, which could be hardly achieved on a membrane with a bigger porous. This device, at the same time, facilitated the establishment of direct cell-to-cell interactions between the epithelial cells and the culture medium in the lower channel, or even with the endothelial cells seeded on the opposing side of the membrane. These features as a whole compound a complete model of epithelial tissue (in this case intestinal, but that can be simply modified by changing the cell line employed), present in different organs of the human body, such as the digestive tube itself, the trachea or the excretory system.

In addition, this work proposed a novel model for brain tumor cells migration towards the extracellular matrix, and, eventually, to the blood vessels. Spheroids from U-87 MG cells (human glioblastoma) were integrated inside a collagen hydrogel, hence recreating the microenvironment of brain tumors. This model can provide insights into the interactions between tumor cells and the surrounding stroma, as well as into the mechanisms regarding glioblastoma progression and metastasis initiation. Furthermore, more solid tumors that lead to metastasis could be modeled with this device, since the pore present in the membrane permits the access of tumor cells into the blood circulation. Some examples could be breast cancer, lung cancer, or prostate, among others; just by substituting the U-87 MG cell line by a tumor cell line proceeding from the type of cancer of interest.

Furthermore, this study presented a multicellular approach to model the system composed of an epithelial monolayer and the supportive connective tissue lying underneath. This was accomplished by embedding fibroblasts (the main responsible cells for the generation of extracellular matrix in the human body) within a collagen hydrogel, and seeding epithelial cells (in this case Caco-2, from human colorectal adenocarcinoma) on the surface of this matrix. This kind of system can be found in many parts of the body, such as the skin or the serous and mucous membranes.”

Minor comments

  1. Proof read the document to ensure grammatical errors

As recommended by the reviewer, the entire manuscript has been thoroughly reviewed to rectify any grammatical errors present within it.

  1. Sub sections can be organized in a better way

The subsections of the manuscript have been reviewed, and efforts have been made to enhance them by incorporating a subsection within the results to accommodate the cellular experiments titled "3.2. Validation of the microfluidic devices," and additionally providing numerical labeling for the various subsections within this section: "3.2.1. Epithelium generation," "3.2.2. Migration assays," and "3.2.3. Coculture assays with epithelial cells and fibroblasts."

  1. Line 53 cell-to cell sentence is not complete. Please check at other locations too.

Due to the reviewer's suggestion, the word "contact" has been added to this sentence.

Reviewer 4 Report

Comments and Suggestions for Authors

This paper reported an MPS device incorporating larger pore sizes to promote cell-cell contacts. The major issue I have with the current results is the lack of detailed characterization backing up the authors’ claims that this design could achieve better interactions between seeded tissues.

Major concerns,

1)      Conventional porous membranes (PET or PC) in MPS devices have small pores (0.45 up to 10 µm in diameter) and relatively low density. However, their thickness is around 10 µm, which is around 10 times thinner than the hydrogel layer used in the current device. It would greatly help the readers to appreciate the advantage if the authors can do some permeability coefficient study, or simulation to validate their claims.

2)  Similarly, it is hard to interpret the current results without a proper control group. Would migration happen slower on traditional membranes? If so, how much slower? Would that have any biological meaning?

3)      As for device design, there have been some attempts to use micropillars/gaps between parallel channels (often with some help of hydrogels) to promote tissue interactions; or use dehydrated ECM as a scaffold altogether. The current introduction needs further updates to reflect the state-of-the-art.

Minor concerns,

(1)    Since this journal is called “biomimetics”, it would be helpful for the authors to clarify what organs/functions those two cell culture studies are trying to mimic, and what advantages they showcased compared to conventional methods.

(2)    Looks like no blue (DAPI) channel was used, as nuclei staining is in cy5. Is it because of the autofluorescence issue of the material? If so, maybe it is worth putting into the discussion.

(3)    Due to the bonding process of 3d printed walls with resin, would there be a dimension limit for the microchannel design? Have the authors check the bonding strength of such a device?

Author Response

We extend our sincere appreciation to the Editor and Reviewers for investing their time in evaluating our submitted manuscript. We have diligently addressed all raised concerns and integrated requisite revisions following the feedback received. In the ensuing sections, you will find the Reviewers' comments highlighted in black, while our responses are presented in italic format.

In response to the comments made by the reviewer 4:

This paper reported an MPS device incorporating larger pore sizes to promote cell-cell contacts. The major issue I have with the current results is the lack of detailed characterization backing up the authors’ claims that this design could achieve better interactions between seeded tissues.

Major concerns,

1)      Conventional porous membranes (PET or PC) in MPS devices have small pores (0.45 up to 10 µm in diameter) and relatively low density. However, their thickness is around 10 µm, which is around 10 times thinner than the hydrogel layer used in the current device. It would greatly help the readers to appreciate the advantage if the authors can do some permeability coefficient study, or simulation to validate their claims.

As discussed in reviewer 1's response 1 and reviewer 3's response 1, experiments were conducted previously using a PC membrane, which revealed no significant evolution in cell migration between compartments. Nevertheless, as recommended by the reviewer, conducting a comparative study on the permeability of the presented membranes against those documented in the literature, such as PC or PET membranes, would be pivotal for this work. However, this would require additional time as setting up the experiment is long.

2)  Similarly, it is hard to interpret the current results without a proper control group. Would migration happen slower on traditional membranes? If so, how much slower? Would that have any biological meaning?

In this case, the discussion in the previous response is relevant, as explained, this work initially sought more porous membranes to allow for increased contact between cavities based on experiments conducted previously, as depicted in Figure A of this document. We concur that conducting a trial with a control (such as a PC or PET membrane) would add value to our study.

3)      As for device design, there have been some attempts to use micropillars/gaps between parallel channels (often with some help of hydrogels) to promote tissue interactions; or use dehydrated ECM as a scaffold altogether. The current introduction needs further updates to reflect the state-of-the-art.

Following the reviewer's advice, these concepts have been introduced in the text by mentioning different types of devices and techniques in which inert material is dispensed between compartments. It can be found in the Discussion section:

“In the field of microfluidics, compartmentalized models are prevalent, typically characterized by material barriers separating individual compartments. While exceptions exist, such as certain devices facilitating total contact between channels [51,52,53], these instances do not align with the model configurations under discussion. Notably, existing models are predominantly structured in a horizontal plane rather than a vertical orientation, thus diverging from the design principles inherent in our proposed models. This distinctive vertical orientation holds significant advantages, facilitating the creation of multiple layers of biological material, thereby enabling the simulation of complex multilayered structures such as the skin.”

[51]         C. Olaizola-Rodrigo et al., “Tuneable hydrogel patterns in pillarless microfluidic devices,” Lab Chip, 2024, doi: 10.1039/d3lc01082a.

[52]         H. Ehlers et al., “Vascular inflammation on a chip: A scalable platform for trans-endothelial electrical resistance and immune cell migration,” Front. Immunol., vol. 14, no. January, pp. 1–11, 2023, doi: 10.3389/fimmu.2023.1118624.

[53]         N. Gjorevski et al., “Tissue geometry drives deterministic organoid patterning,” Science (80-. )., vol. 375, no. 6576, 2022, doi: 10.1126/science.aaw9021.

Minor concerns,

(1)    Since this journal is called “biomimetics”, it would be helpful for the authors to clarify what organs/functions those two cell culture studies are trying to mimic, and what advantages they showcased compared to conventional methods.

Thanks to the reviewer's suggestion, we have incorporated additional applications of the device beyond those previously mentioned (reviewer 3, second question), and this is incorporated into the Discussion section of the manuscript:

“Firstly, epithelial tissue was modeled using the Mesh device. By seeding HCT-116 (human colon carcinoma) cells on the microfluidic device membrane, we could mimic the characteristics of the intestinal epithelium. To accomplish this, the utilization of an intermediate porous membrane, such as the one present in the Mesh device, allowed the formation of a monolayer, which could be hardly achieved on a membrane with a bigger porous. This device, at the same time, facilitated the establishment of direct cell-to-cell interactions between the epithelial cells and the culture medium in the lower channel, or even with the endothelial cells seeded on the opposing side of the membrane. These features as a whole compound a complete model of epithelial tissue (in this case intestinal, but that can be simply modified by changing the cell line employed), present in different organs of the human body, such as the digestive tube itself, the trachea or the excretory system.

In addition, this work proposed a novel model for brain tumor cells migration towards the extracellular matrix, and, eventually, to the blood vessels. Spheroids from U-87 MG cells (human glioblastoma) were integrated inside a collagen hydrogel, hence recreating the microenvironment of brain tumors. This model can provide insights into the interactions between tumor cells and the surrounding stroma, as well as into the mechanisms regarding glioblastoma progression and metastasis initiation. Furthermore, more solid tumors that lead to metastasis could be modeled with this device, since the pore present in the membrane permits the access of tumor cells into the blood circulation. Some examples could be breast cancer, lung cancer, or prostate, among others; just by substituting the U-87 MG cell line by a tumor cell line proceeding from the type of cancer of interest.

Furthermore, this study presented a multicellular approach to model the system composed of an epithelial monolayer and the supportive connective tissue lying underneath. This was accomplished by embedding fibroblasts (the main responsible cells for the generation of extracellular matrix in the human body) within a collagen hydrogel, and seeding epithelial cells (in this case Caco-2, from human colorectal adenocarcinoma) on the surface of this matrix. This kind of system can be found in many parts of the body, such as the skin or the serous and mucous membranes.”

When contrasting the outcomes of these experiments against those utilizing membranes documented in current literature (e.g., PC, PET, PCL...), a notable advantage lies in the increased contact area between distinct compartments. This characteristic facilitates biomimicry, as it closely approximates real models of epithelial tissue or tumor cell migration towards extracellular matrix or blood vessels, akin to our experimental setup.

(2)    Looks like no blue (DAPI) channel was used, as nuclei staining is in cy5. Is it because of the autofluorescence issue of the material? If so, maybe it is worth putting into the discussion.

Thank you for your question. We did not use the blue pseudocolor for visualizing nuclei because it did not provide enough contrast with the one used for ZO-1 staining (red). Instead, we decided to use cyan pseudocolor. An illustrative example of this is shown in the following image:

(the figure E can be seen in the document attached to this response.)

Figure E: Nuclei visualization of the epithelium. Tight junctions with ZO-1 (red) and nuclei counter labeling with Hoechst (blue/cyan). Scale bar: 100 µm.

(3)    Due to the bonding process of 3d printed walls with resin, would there be a dimension limit for the microchannel design? Have the authors check the bonding strength of such a device?

The 3d printed part comprises supplementary wells integrated into the injection unit, thereby elevating the well's depth to accommodate a larger volume of medium, facilitating prolonged experimental durations. This augmentation does not constrain the dimensions of the channels, as they remain separate entities in terms of manufacturing.

Round 2

Reviewer 1 Report

Comments and Suggestions for Authors

None

Author Response

No comments have been received from this reviewer.

Reviewer 4 Report

Comments and Suggestions for Authors

I appreciate the author's efforts in addressing some of my previous concerns. The responses to my primary major concern 1) and 2) were quite non-responsive. The current updates didn't improve the paper quality in a publishable state.

Author Response

The reviewer's feedback has been included in the attached file.

Round 3

Reviewer 4 Report

Comments and Suggestions for Authors

I appreciate the authors' efforts in providing new experimental results. I have some minor concerns listed as follows:

1) Would there be any concerns that even the 4kD FIT-Dextran cannot diffuse through the hydrogel/cell layer enough (to give any signal in the plate reader) in 3 hours?

2) Instead of using a plate reader, will there be any measurable signal via fluorescent microscopy along the z-axis? 

3) Based on the permeability coefficient calculation, is this fabricated microenvironment comparable to the e.g., tumor microenvironment in vivo?
